# Dynamic Impact of the Perceived Value of Public on Panic Buying Behavior during COVID-19

Qing-Hua Mao 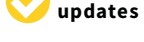, Jiang-Xiao Hou and Peng-Zhen Xie *

School of Economics and Management, Yanshan University, Qinhuangdao 066004, China;
maoqh@ysu.edu.cn (Q.-H.M.); hou150216@163.com (J.-X.H.)
* Correspondence: xiepengzhen123@163.com

**Abstract:** In this paper, an evolutionary game model for the development of panic buying events in COVID-19 is constructed by studying the dynamic process of the public and the government adjusting their strategic choices and playing a continuous game. This paper uses regret theory to depict the public's perceived value of the items in the panic buying situation, describes the characteristics of each stage of the rumors spreading process and the evolution process of panic buying events, and introduces the variable of public critical ability to measure the public's panic buying willingness. The results show that the government's intervention measures according to the characteristics of different stages can effectively control the continuous fermentation time and influence the scope of panic buying events. The implementation of the government's rumor-refutation strategy will also significantly affect the volume of public panic buying, which will help the government timely understand the public's epidemic prevention needs in COVID-19, relieve public panic, and provide a basis for the effective management and scheduling of emergency supplies.

**Keywords:** panic buying event; evolutionary game; regret theory; rumor spreading process theory; COVID-19



## 1. Introduction

When COVID-19 occurs, the public's unknown and uncertain information about the transmission and prevention methods of the virus often leads to rampant rumors, triggers a series of panic buying behavior, and has a significant undesirable impact on society [1]. Panic buying is defined as a phenomenon of a sudden increase in buying of one or more essential goods in exceeds of regular need provoked by adversity, usually a disaster or an outbreak resulting in an imbalance between supply and demand [2]. Panic buying generally exists during public emergencies. In different situations, consumers will stock different items depending on different needs. For example, during influenza, consumers will buy flu control drugs in bulk; in the case of water contamination, they will hoard drinking water; in the case of avian flu, they will rush to buy condiments; in the case of a nuclear leak, they will rush to buy iodized salt, and so on. As the COVID-19 epidemic is extremely easy to spread and infect, there is a serious shortage of medical emergency supplies such as masks, protective clothing, and protective masks, and consumers are also in dire need of large quantities of masks and disinfectant water to meet their basic epidemic prevention needs.

At present, the research of panic buying events in public health emergencies mainly focuses on the factors influencing the events and the analysis of the evolution law. In the research on the influencing factors of panic buying events, some scholars point out that the amount of panic buying is considered to depend on perceived need, and the sudden spread of rumors and misinformation will lead to the development of people's negative emotions [3], and the public's increased risk perception and trust in social media deepened their willingness to wait in line for purchase [4]. Risk factors, protective factors [5],

and external stimulus factors [6] can also affect consumers' moods and impulse buying behavior. Vilk et al. [7] first showed that social media users' herd mentality differs in a crisis. Motamed et al. [8] also divided factors influencing panic buying into six categories, including cognitive, emotional, behavioral, social, and economic factors, as well as government action. In the panic buying process, the normal value of panic buying items is replaced by the perceived value of consumers, and its price is artificially pushed up. Perceived value refers to the overall assessment of the utility of panic buying items after weighing the perceived benefit against the cost to the consumers in the panic buying. However, few studies have considered the public's own perceived value and the choices that the public may make in the context of loss avoidance so as to avoid regret as much as possible and ensure minimal loss. In the research of the evolution law of panic buying events, Lindenmeier et al. [9] used an affective and a cognitive effect channel to explain panic buying. Dulam et al. [10] have developed an agent-based model that can simulate the various outcomes of a crisis using a consumer panic buying model and a supply chain model, and the model quantitatively evaluates the panic purchase intention of a consumer. Wang et al. [11] defined the roles and game relations of participants in sudden panic buying events and analyzed the game equilibrium results of different evolutionary stages of panic buying events. Dai et al. [12] examined how user-generated anti-panic buying messages online could be leveraged to combat panic buying. Niu et al. [13] suggested that targeted information on the development of the epidemic should be provided to different groups of people to reduce their fear of the epidemic, improve the security and supply mechanism of emergency goods, and maintain the price stability of protective goods to avoid hoarding of emergency goods. Zhu et al. [14] applied the theory of propagation model to reveal the existence of the equilibrium points, the backward bifurcation, and the local stability so as to reduce the optimal control of rumor propagation frequency. Li et al. [15] combined the individual decision model and the panic transmission model in an uncertain environment to study the panic buying behavior caused by online rumors. The above studies only correlate rumors with price changes of panic buying items but fail to study the interaction between the rumors spreading process and panic buying events.

In view of this, taking the COVID-19 as the background, this paper uses regret theory to depict the public's perceived value of the panic buying items in the process of panic buying, describes the characteristics of each stage of the rumors spreading process, and the evolution process of panic buying events, and introduces the variable of public critical ability to measure the public's panic buying willingness so as to further optimize the government's crisis control ability and promote the governance research of group panic buying behavior in major public health emergencies.

## 2. Theory Overview and Problem Description

### 2.1. Regret Theory

The core idea of regret theory is that the decision makers will compare their current situation with the situation they might be in. If the decision makers find that they can achieve a better result by choosing other alternatives, they may feel regret in their hearts. On the contrary, the decision makers will feel joy. Regret theory has the following three core characteristics: (1) it considers both the risk attitude and the regret attitude of the decision makers; (2) it quantifies the psychological emotions of regret of the decision makers; (3) it involves few parameters and its calculation process is easy to understand [16]. Decision models are usually bounded rational because of time pressure, limited experiences, information overload, or insufficient. Regret theory is more simple than the other ones in real decision-making applications [17].

In the decision-making process, the decision makers will anticipate the possible regret or joy of the decision and try to avoid choosing the plan that will make them regret it. This may probably link to the gain and loss (benefits and costs). When consumers feel gains, they feel joyful. However, they feel remorseful or regret when they probably make a decision that loss is greater than utility/benefits [18,19]. That is, the decision makers are

regret avoidant. According to regret theory, the decision maker's perceived utility function is composed of two parts: the utility function and the regret function about the current outcome. Let $x$ and $y$ represent the results obtained by choosing plan $A$ and $B$ respectively, and the perceived utility of the decision makers for plan $A$ is:

$$u(x,y) = v(x) + R(v(x) - v(y)) \tag{1}$$

where $v(x)$ and $v(y)$ represent the utility obtained by the decision makers from plan $A$ and plan $B$, respectively. $R(v(x) - v(y))$ represents the regret and joy value. When $R(v(x) - v(y)) > 0$, $R(v(x) - v(y))$ is the joy value, indicating that the decision makers are joyful to choose plan $A$ over plan $B$; When $R(v(x) - v(y)) < 0$, $R(v(x) - v(y))$ is the regret value, indicating that the decision makers are regretful to choose plan $A$ over plan $B$.

### 2.2. Problem Description

The rumors spreading can give a specific item a specific utility in a specific time and space, and the rumors are often accompanied by sudden panic buying events. For example, there are rumors that wearing a mask is not contagious and shuanghuanglian oral liquid (It is an over-the-counter medicine with the ability to clear heat and detoxify and improve the immune function of the body) can effectively suppress the virus during the COVID-19. Driven by rumors, the price of a particular item will go up, the utility value of that item will be magnified, and the public psyche will have a higher expected value for that item. Panic buying behavior occurs when a majority of the public buys an item in a group. In the process of panic buying, the public may feel joy because the demand for participating in panic buying has been satisfied, or they may feel regret that they have paid a higher cost by participating in the panic buying. In the different stages of a panic buying event, the price of the snapped items may fluctuate dramatically, and there is a possibility that the public may blindly purchase an excessive amount of the snapped items at a much higher price than normal. These snapped items may be significantly reduced in price after the point when they are snapped up or may remain in large surplus until the public emergency subsides, resulting in some loss to the public. This paper intends to apply regret theory to public panic buying behavior to depict the public's perceived value of panic buying items. While the public is panic buying, the spread of relevant rumors will boost the development of panic buying events from both behavioral and psychological aspects. Some studies have classified the evolution stage of public opinion [20,21] and the stage of panic buying events [22,23] into four stages: the germination stage, the intensification stage, the mitigation stage, and the regression stage. However, there is a certain interaction between the evolution of public opinion and the evolution of panic buying events, in that the spread of public opinion will promote the development of panic buying events, and the development of panic buying events will, in turn, boost the spread of public opinion, while existing studies have not considered the interaction between the two. On the basis of the above research, this paper intends to combine the panic buying behavior with the rumors propagation process and describes the rumors propagation characteristics of the panic buying process in the germination stage, intensification stage, mitigation stage, and regression stage.

In the process of panic buying, due to the impact of uncertain conditions, the regret and joy of the public participating or not participating in panic buying at different stages are different. In the process of rumor spreading, governments at all levels may introduce policies to suppress the development of rumors or ignore them after weighing the cost of rumor dispelling, the scope of impact, and the rewards and punishments of higher governments. At the same time, the emotion and position orientation of the rumors themselves have become a powerful driver of rumors spreading by the public, which also makes the occurrence of panic buying behavior often inseparable from rumors spreading.

It is assumed that the public's behavior is independent of each other in the panic buying process, and they will decide their behavior according to the strategy of least regret

decision to avoid regret. Therefore, the expected utility function $v(x)$ is a monotonically increasing concave function expressed by a power function. That is:

$$v(x) = x^{\alpha} \tag{2}$$

where $\alpha$ is the regret avoidance coefficient of the public in panic buying events, $0 < \alpha < 1$, $v(x)$ satisfies $v'(x) > 0$, $v''(x) < 0$.

## 3. Evolutionary Game Analysis of Panic Buying Events

### 3.1. Model Hypothesis

According to the performance of multiple participants in panic buying events, the phenomenon of repeated panic buying by the public and repeated refuting by the government usually exists. The occurrence and evolution of panic buying events are essentially a dynamic process in which the public and the government adjust their strategic choices and play a continuous game. In order to do this, the following hypothesis is put forward:

**Hypothesis 1 (H1).** *The government and the public participating in the game are rational economic people, that is, to maximize their own interests as the goal.*

**Hypothesis 2 (H2).** *As panic buying events develop, the public and the government will adjust their strategies according to their own perceptions.*

**Hypothesis 3 (H3).** *The public's gains and losses will be measured by the public's perceived value, purchase willingness, and purchase volume of panic buying items.*

**Hypothesis 4 (H4).** *In the process of panic buying, the price of panic buying items is higher than the normal price, and the panic buying volume is higher than the normal demand.*

**Hypothesis 5 (H5).** *The combined costs to the government include the cost of refuting rumors, social losses, and rewards or penalties from higher governments.*

### 3.2. The Critical Ability of Public

As the information environment around them changes, the public's cognitive ability and ability to distinguish between true and false information will gradually increase, and they will have the ability to refute and deny ideas or actions that are perceived to be wrong, which is called critical ability.

In the research of rumors transmission, the rumors transmission model proposed by American social psychologists Allport and Postman is widely used, that is $R = I * A$. Where $R$ represents the influence of rumors, $I$ represents the importance of rumors to relevant people, and $A$ represents the vagueness of rumors. On this basis, Kross added the element of the critical ability of the public, that is $R = I * A/C$.

The fuzziness of major emergencies is an important factor influencing the generation and dissemination of rumors. The more important events are considered by the public, the more ambiguous they feel, and the faster and wider the spread of rumors. The weaker the critical ability of the public is, the greater the volume of rumors spreading [24]. The stronger the critical awareness of the public is, the intensity of rumors spreading will gradually decrease, and the negative impact range brought by rumors spreading will also decrease [25]. This paper intends to introduce the variable of the critical ability of the public to depict the willingness of the public's panic buying. The stronger the critical ability of the public is, the lower the trust in rumors and the lower willingness to participate in panic buying. There is an inverse relationship between the critical ability of the public and the willingness to panic buying. That is:

$$f(u) = \frac{\varphi_i}{u} \tag{3}$$

where $u$ is the critical ability of the public to rumors. When the public participates in panic buying, $\varphi_i = 1$. When the public does not participate in panic buying, $\varphi_i = 0$.

### 3.3. Model Building

Based on the assumptions of the government's strategies and the public's strategies, four different game strategy combinations will appear during the interaction between these two subjects, namely:

(1)  In the germination stage of panic buying events, the game strategy combination is that the public does not participate in panic buying, and the government does not refute the rumors.

(2)  In the intensification stage of panic buying events, the game strategy combination is that the public participates in panic buying, and the government does not refute the rumors.

(3)  In the mitigation stage of panic buying events, the game strategy combination is that the public participates in panic buying, and the government does not refute the rumors.

(4)  In the regression stage of panic buying events, the game strategy combination is that the public participates in panic buying, and the government refutes the rumors.

The specific circumstances of each stage are described below.

(1)  In the germination stage of panic buying events, information related to the epidemic was released slowly, authoritative information was missing, and the public made their own speculations and had low information judgment literacy. The number of online rumors began to rise slowly and spread easily in a short time. Some potential factors that may induce large-scale panic buying are accumulating. If they are not highly valued or effectively solved, it may lead to further deterioration of the situation. However, in this stage, the spread scope of rumors about potential panic buying items is small, and the critical ability of the public to rumors and perceived value of panic buying items are low. Most of the public will still consume rationally, and they are not prone to panic buying behavior. The normal purchase volume is $m$, and the perceived value is $mp + R(v(x) - v(y))$. At the same time, if the government chooses to refute rumors, it may not only pay the cost of refuting rumors but also cause unnecessary panic due to premature attention. In this case, the cost to be paid by the government is 0.

(2)  In the intensification stage of panic buying events, the public's critical consciousness began to improve gradually by popularizing the knowledge related to emergencies and emergency management and providing an open and effective platform for information inquiry and exchange. However, there was a sharp rise in the number of online rumors about the epidemic, which triggered the collective panic emotions of the public. The number of panic buying individuals has increased rapidly, the group scale has expanded sharply, and the relationship between supply and demand of panic buying items has been abnormal. In this stage, the spread of rumors gradually expanded, and the public's trust in rumors and the perceived value of panic buying items increased, which promoted their panic buying willingness and behavior. The public's panic buying willingness is $f(u)$, the purchase volume is $m_0$ and the perceived value is $m_0 p_0 + R(v(y) - v(x))$. At the same time, if the government insists on refuting the rumors, it will increase its own losses. The government has the motivation to choose not to refute the rumors in order to calm the incident on its own.

(3)  In the mitigation stage of panic buying events, the number of rumors about the development of the epidemic has decreased significantly due to the stable development of the epidemic, and the rumors are more from information misreading or subjective speculation. The government actively announced the epidemic information through press conferences and mainstream media to satisfy the public's right to know and answered some rumors through official platforms and experts to minimize the ambiguity of information and curb the breeding of rumors. In this stage, the government actively

refuted rumors, which reduced the public's perceived value of panic buying items, improved their critical ability, and gradually lost their willingness to panic buying. The price of panic buying items rose, but part of the public returned to rationality and gave up participating in panic buying. Its perceived value is $mp_0 + R(v(x) - v(y))$. At the same time, if the government does not refute the rumors, the social loss caused by panic buying events will be difficult to control. In this case, the government will pay a certain cost $c$ for refuting rumors and will be rewarded by higher government $d$ for the refuting rumors work.

(4) In the regression stage of panic buying events, the government continued to release epidemic information openly and transparently, and the epidemic was effectively controlled. Online rumors about the epidemic gradually retreated steadily, and mainstream media effectively guided public opinion and controlled the development pace of public opinion. The public's comprehensive literacy and ability to distinguish the authenticity of information are improved, the number of individuals involved in panic buying is significantly reduced, and panic buying behavior is basically calmed down. In this stage, the government keeps refuting rumors, and the public has strong critical consciousness. However, they still need to participate in the panic buying to meet their own normal epidemic prevention needs. Therefore, the public will increase the purchased quantity due to the decrease in the price of panic buying items and will still have the panic buying willingness $f(u)$. The purchase quantity of panic buying items is $m_0$ and the perceived value is $m_0 p + R(v(y) - v(x))$. At the same time, the cost of refuting rumors caused by the government is $c$, the social loss caused by the government is $\omega$, and the reward for effectively guiding public opinion received by the higher government is $d$.

To sum up, the payment matrix of both sides of the game is constructed in Table 1.

**Table 1.** Payment matrix.

| Strategy Choice | | Public | |
|---|---|---|---|
| | | Participate in Panic Buying ($y$) | Not Participate in Panic Buying ($1-y$) |
| Government | refute the rumors ($x$) | $(-\omega - c + d, m_0 p + R(y^\alpha - x^\alpha) + f(u))$ | $(-c + d, mp_0 + R(x^\alpha - y^\alpha))$ |
| | not refute the rumors ($1 - x$) | $(-\omega - d, m_0 p_0 + R(y^\alpha - x^\alpha) + f(u))$ | $(0, mp + R(x^\alpha - y^\alpha))$ |

Suppose that the probability of the government choosing to refute the rumors is $x$, and the probability of not refuting the rumors is $1 - x$; The probability of the public participating in panic buying is $y$, and the probability of not participating in panic buying is $1 - y$.

According to the payment matrix of the evolutionary game, the utility of the government refutes the rumors is:

$$U_{A1} = y(-\omega - c + d) + (1 - y)(-c + d) \tag{4}$$

The utility of not refuting the rumors is:

$$U_{A2} = y(-\omega - d) \tag{5}$$

The average utility is:

$$\overline{U}_A = xU_{A1} + (1 - x)U_{A2} \tag{6}$$

Similarly, when the public participates in panic buying, the perceived value of the items is:

$$U_{B1} = x[m_0 p + R(y^\alpha - x^\alpha) + f(u)] + (1 - x)[m_0 p_0 + R(y^\alpha - x^\alpha) + f(u)] \tag{7}$$

When the public does not participate in panic buying, the perceived value of the items is:

$$U_{B2} = x[mp_0 + R(x^\alpha - y^\alpha)] + (1-x)[mp + R(x^\alpha - y^\alpha)] \tag{8}$$

The average perceived value is:

$$\overline{U}_B = yU_{B1} + (1-y)U_{B2} \tag{9}$$

Based on the above analysis, duplicate dynamic equations of the evolutionary game between government and public are constructed, respectively:

$$F_A(x) = \frac{dx}{dt} = x(1-x)(dy - c + d) \tag{10}$$

$$F_B(y) = \frac{dy}{dt} = y(1-y)[x(m_0p - mp_0 - m_0p_0 + mp) + m_0p_0 - mp + u_1 - u_2 + f(u)] \tag{11}$$

Among them, $u_1$ is the regret-joy value when the public participates in panic buying, $u_1 = R(y^\alpha - x^\alpha)$. $u_2$ is the regret-joy value when the public does not participate in panic buying, $u_2 = R(x^\alpha - y^\alpha)$.

Simultaneous equations, let $F_A(x) = 0$, $F_B(y) = 0$. There are five equilibrium points, namely (0,0), (0,1), (1,0), (1,1), $(x^*, y^*)$, which $x^* = \frac{mp - m_0p_0 - u_1 + u_2 - f(u)}{m_0p - mp_0 - m_0p_0 + mp}$, $y^* = \frac{c-d}{d}$.

According to the judgment method of evolutionary stability strategy (ESS) [26], the Jacobi matrix can be obtained as:

$$J = \begin{bmatrix} \frac{\partial F_A(x)}{\partial x} & \frac{\partial F_A(x)}{\partial y} \\ \frac{\partial F_B(y)}{\partial x} & \frac{\partial F_B(y)}{\partial y} \end{bmatrix}$$

$$= \begin{bmatrix} (1-2x)(dy - c + d) & x(1-x)d \\ y(1-y)(m_0p - mp_0 - m_0p_0 + mp) & (1-2y)\begin{bmatrix} x(m_0p - mp_0 - m_0p_0 + mp) \\ + m_0p_0 - mp + u_1 - u_2 + f(u) \end{bmatrix} \end{bmatrix} \tag{12}$$

The equilibrium point is ESS if and only if $Det(J) > 0$ and $Tr(J) < 0$.

### 3.4. Model Analysis

This paper further explores the local stability of each equilibrium point according to the change of the determinant of the Jacobi matrix and the sign of the trace under different conditions.

(1) When $(-c + d) < 0$, $(2d - c) < 0$, $[m_0p_0 - mp + u_1 - u_2 + f(u)] < 0$ and $(m_0p - mp_0 - m_0p_0 + mp) < 0$, the system has five equilibrium points, including three unstable points, a locally asymptotically stable point, and a saddle point. $E_1(0,0)$ for a saddle point, namely the strategy combinations (The public does not participate in panic buying. The government does not refute the rumors). The practical significance of this evolutionary stabilization strategy is that the regret value of the public not participating in panic buying is the least, and the benefit is higher than participating in panic buying. This situation appeared mostly in the germination period of panic buying events; the government should strengthen the real-time monitoring of the rumors spreading situation and the public panic buying signs so as to make adequate preparations for timely policy adjustment. The determinant of the equilibrium point, trace, and stability are shown in Table 2.

(2) When $(2d - c) < 0$, $(-c + d) < 0$, $[m_0p_0 - mp + u_1 - u_2 + f(u)] > 0$ and $(m_0p - mp_0 - m_0p_0 + mp) > 0$, the system has five equilibrium points, including three unstable points, a locally asymptotically stable point, and a saddle point. $E_2(0,1)$ for a saddle point, namely the strategy combinations (The public participates in panic buying. The government does not refute the rumors). The practical significance of this evolutionary stabilization strategy is that the public will be delighted to panic buy what they need, and the degree of regret is low. This situation appeared mostly

during the intensification period of panic buying events; in order to encourage the government to actively perform its duties, the higher government should increase the punishment for its dereliction of duty and urge the government to adjust product production and balance product prices. The determinant of the equilibrium point, trace, and stability are shown in Table 3.

(3) When $(-c + d) > 0$, $(2d - c) > 0$, $[m_0 p_0 - mp + u_1 - u_2 + f(u)] < 0$ and $(m_0 p - mp_0 - m_0 p_0 + mp) < 0$, the system has five equilibrium points, including three unstable points, a locally asymptotically stable point, and a saddle point. $E_3(1, 0)$ for a saddle point, namely the strategy combinations (The public does not participate in panic buying. The government refutes the rumors). The practical significance of this evolutionary stabilization strategy is that the price of panic buying items rises, and the public will be glad that the loss caused by not participating in panic buying is small. This situation appeared mostly in the mitigation period of panic buying events; the government should pay attention to repeatedly refuting the rumors and improving the intensity of refuting the rumors, strengthening the openness and transparency of information, reducing the public's trust in rumors, and advocating reasonable public consumption. The determinant of the equilibrium point, trace, and stability are shown in Table 4.

(4) When $(2d - c) > 0$, $(-c + d) > 0$, $[m_0 p_0 - mp + u_1 - u_2 + f(u)] > 0$ and $(m_0 p - mp_0 - m_0 p_0 + mp) > 0$, the system has five equilibrium points, including three unstable points, a locally asymptotically stable point, and a saddle point. $E_4(1, 1)$. for a saddle point, namely the strategy combinations (the public participates in panic buying. The government refutes the rumors). The practical significance of this evolutionary stabilization strategy is that the public will be glad that the price of items will fall and the demand will be satisfied. This situation appeared mostly in the regression period of panic buying events; in order to curb the occurrence of panic buying behavior, the government should continue to promote refuting of the rumors, timely notify the market supply of items and the disposal progress of panic buying behavior, and enhance the public's awareness of loss avoidance. The determinant of the equilibrium point, trace, and stability are shown in Table 5.

(5) When $0 < x^* < 1$ and $0 < y^* < 1$, this paper substitutes the equilibrium point $E_5(x^*, y^*)$ into the Jacobi matrix, and the Jacobi matrix of this system is as follows:

$$J = \begin{bmatrix} 0 & \frac{[mp - m_0 p_0 - u_1 + u_2 - f(u)][m_0 p - mp_0 + u_1 - u_2 + f(u)]d}{(m_0 p - mp_0 - m_0 p_0 + mp)^2} \\ \frac{(c-d)(2d-c)(m_0 p - mp_0 - m_0 p_0 + mp)}{d^2} & 0 \end{bmatrix} \quad (13)$$

Among them, $Det(J) < 0$ and $Tr(J) = 0$. Therefore, the equilibrium point $E_5(x^*, y^*)$ is not the asymptotically stable point of the system but the saddle point of the system. From the local stability analysis of the above evolutionary game, the following inferences can be drawn: ① When $y = y^*$, there is always $F_A(x) = 0$; that is, the value of $x$ is within the defined range, and the system eventually will reach an evolutionary stable state. When the public participates in panic buying behavior, no matter whether the government introduces a policy of disinformation or not, there is no difference in the value of the public's profit. When $x = x^*$, there is always $F_B(y) = 0$; that is, the value of $y$ is within the defined range, and the system eventually will reach an evolutionary stable state. When the government adopts a disinformation strategy, no matter whether the public participates in the rush or not, there is no difference in the value of the government's profit. ② When $y > y^*$, $dy - c + d > 0$, at this time, $x = 0$ and $x = 1$ are two possible stable points. When $x = 0$, $\frac{\partial F_A(x)}{\partial x} = 1 > 0$; and when $x = 1$, $\frac{\partial F_A(x)}{\partial x} = -1 < 0$. Therefore, the evolutionary game will reach a stable state, and $x = 1$ is the only possible stable point; that is, the strategy of the government election will gradually shift from no disinformation to disinformation, and adopting a disinformation strategy will eventually become the evolutionary stability of the government strategy. In the same way, when $y < y^*$, $x = 0$ is the only possible stable point; that is, the strategy of the government election gradually shifts from disinformation to no disinformation, and the strategy of not adopting disinformation strategy will

eventually become the evolution of the government's stable strategy. ③ When $x > x^*$, $x(m_0 p - m p_0 - m_0 p_0 + mp) + m_0 p_0 - mp + u_1 - u_2 + f(u) > 0$, at this time, $y = 0$ and $y = 1$ are two possible stable points. When $y = 0$, $\frac{\partial F_B(y)}{\partial y} = 1 > 0$; and when $y = 1$, $\frac{\partial F_B(y)}{\partial y} = -1 < 0$. Therefore, the evolutionary game will reach a stable state, and $y = 1$ is the only possible stable point; that is, the strategy of the public election will gradually shift from not participating in panic buying to participating in panic buying, and the participating in panic buying strategy will eventually become the evolutionary stability of the public strategy. In the same way, when $x < x^*$, $y = 0$ is the only possible stable point; that is, the strategy of the public election gradually shifts from participating in panic buying to not participating in panic buying, and the strategy of the not participating in panic buying strategy will eventually become the evolution of the government's stable strategy.

**Table 2.** Analysis of the local stability of the equilibrium point.

| Equilibrium Point | Det(J) | Tr(J) | Local Stability |
|---|---|---|---|
| $E_1(0,0)$ | + | − | ESS |
| $E_2(0,1)$ | − | +/− | Unstable |
| $E_3(1,0)$ | − | +/− | Unstable |
| $E_4(1,1)$ | + | + | Unstable |
| $E_5(x^*,y^*)$ | − | 0 | Saddle point |

**Table 3.** Analysis of the local stability of the equilibrium point.

| Equilibrium Point | Det(J) | Tr(J) | Local Stability |
|---|---|---|---|
| $E_1(0,0)$ | − | +/− | Unstable |
| $E_2(0,1)$ | + | − | ESS |
| $E_3(1,0)$ | + | + | Unstable |
| $E_4(1,1)$ | − | +/− | Unstable |
| $E_5(x^*,y^*)$ | − | 0 | Saddle point |

**Table 4.** Analysis of the local stability of the equilibrium point.

| Equilibrium Point | Det(J) | Tr(J) | Local Stability |
|---|---|---|---|
| $E_1(0,0)$ | − | +/− | Unstable |
| $E_2(0,1)$ | + | + | Unstable |
| $E_3(1,0)$ | + | − | ESS |
| $E_4(1,1)$ | − | +/− | Unstable |
| $E_5(x^*,y^*)$ | − | 0 | Saddle point |

**Table 5.** Analysis of the local stability of the equilibrium point.

| Equilibrium Point | Det(J) | Tr(J) | Local Stability |
|---|---|---|---|
| $E_1(0,0)$ | + | + | Unstable |
| $E_2(0,1)$ | − | +/− | Unstable |
| $E_3(1,0)$ | − | +/− | Unstable |
| $E_4(1,1)$ | + | − | ESS |
| $E_5(x^*,y^*)$ | − | 0 | Saddle point |

### 3.5. The Simulation Analysis of Panic Buying Masks Behavior

In order to test the validity of the impact of the government's rumors refuting strategy and the critical ability of the public on rumor spreading, this paper will simulate the public's panic buying of masks during COVID-19. Drawing on the game simulation literature [27,28], the game relationship in this paper assumes $c = 30, d = 50, \omega = 100$, $m = 2, p = 2, u = 0.5, u_1 = 0.5, u_2 = 0.5, x = 0.5, y = 0.5$, substitutes the parameters into the replicated dynamic equations set (10) and (11) and uses Matlab to calculate the

differential equations, which can result in the curves of variables $m_0$ and $u$ with time $y$. When examining the impact of the implementation of the government's rumors refuting strategy on the public's panic buying behavior, make $p_0 = 4$ and examine $m_0$; the simulation results are shown in Figures 1 and 2. When examining the impact of the critical ability of the public on public panic buying willingness, make $m_0 = 4$ and $u$ take 0.3, 0.5, 0.7, and 0.9, respectively; the simulation results are shown in Figure 3.

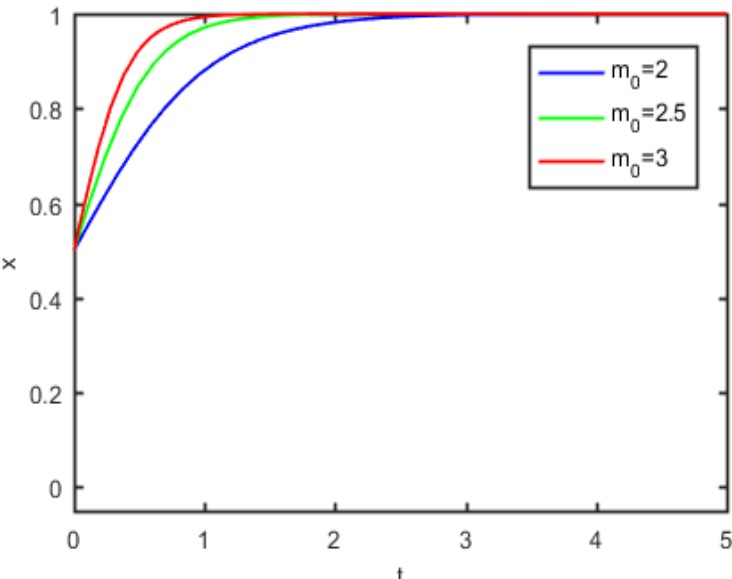

**Figure 1.** The impact of the government not refuting rumors on the volume of public panic buying.

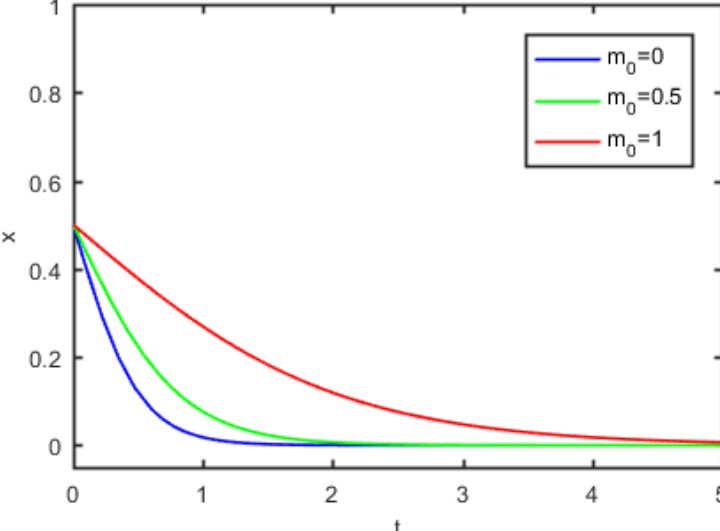

**Figure 2.** The impact of the government refuting rumors on the volume of public panic buying.

As can be seen from Figure 1, when the government did not refute the rumors, the volume of public panic buying gradually evolved to 1, and with the increase in $m_0$, the time for its evolution to 1 gradually shortened. In the germination stage of the epidemic, the public's trust in the rumors and perceived value of masks were low, so they did not immediately realize the seriousness of the epidemic. At this time, the public's demand for masks is growing slowly, and the price of masks is still at a stable level. If the public does not participate in panic buying, it will not bring losses and will not feel regret compared with participating in panic buying. The government also focuses on the investigation and monitoring of public opinion. The result of the game is not refuting rumors and not panic

buying. During the intensification period, the public's trust in rumors and the perceived value of masks have increased rapidly, and the price of masks remains high. Masks in pharmacies, stores, and other platforms have been sold out, and heavy money is hard to find. Although the increase in the price of masks will bring some economic losses to the public, the public will feel joy because they have met their own epidemic prevention needs. At the same time, considering the impact of intervention timing on economic and social benefits, the government did not take an intervention. The result of the game is not refuting rumors and panic buying.

As can be seen from Figure 2, when the government adopts the rumors refuting strategy, it has a significant impact on the volume of public panic buying. When the government refuted the rumors, the volume of public panic buying gradually evolved to 0, and with the increase in $m_0$, the time for its evolution to 0 gradually prolonged. In the mitigation period, the supply of masks is becoming more abundant, and the price shows a downward trend. On the premise that part of the demand has been satisfied, the public will give up participating in panic buying in order to avoid causing losses and feeling regret. In order to avoid the deterioration of the situation, the government is bound to actively take intervention behavior. The result of the game is refuting rumors and not panic buying. In the regression period, the public's demand for masks has dropped, the supply has been sufficient, and the price has dropped to the normal level. The intervention of higher levels of government and the transparency of the epidemic information has prompted the public to return to rationality. At this time, the public will hoard a certain number of masks for reserve due to the drop in the price of masks and ensure the satisfaction of the demand, and the joy of public participation in panic buying is higher than the possible regret. At the same time, the government will actively intervene in order to fulfill the social responsibility of maintaining the market and avoid being punished by the higher government. The result of the game is refuting rumors and panic buying.

In conclusion, in the storm of the masks panic buying, the public rushed to buy masks at a rapid pace to meet their own epidemic prevention needs. When the government successively released relevant rumors refuting the information, the public's demand for information was satisfied, their scientific literacy was improved, they were able to maintain certain rationality in panic buying events, and the purchased quantity of masks gradually decreased. However, when the government has not issued the rumors refuting strategy, the public will increase the purchase of masks due to the impact of rumors spreading and a herd effect.

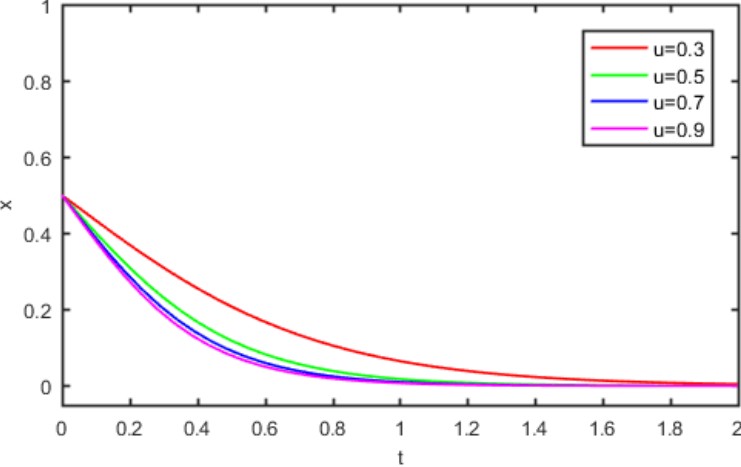

**Figure 3.** The impact of the critical ability of public on public panic buying willingness.

As can be seen from Figure 3, with the improvement of the critical ability of the public, the public's trust in rumors and the willingness to participate in panic buying decreased, gradually evolving to 0. At the beginning of the epidemic, the public did not realize its

seriousness and lacked awareness of epidemic prevention. Therefore, they would believe rumors and participate in panic buying of masks. With the openness and transparency of epidemic related information and the government's active refutation of rumors, the public's comprehensive literacy and ability to distinguish the truth and falsehood of information have gradually improved, and the public's purchase of masks has returned to rationality and gradually lost their willingness to panic buying.

## 4. Discussion

Firstly, this study shows that when taking into account their own perceived value, the public will make different participation decisions according to their own minimum loss in the development of panic buying events. Ma and Liao's [1] study also shows that the link between scarcity and panic buying is transmitted via reduced levels of perceived control and enhanced levels of panic, especially in life-threatening crises such as public health emergencies. That is, there is a direct link between perceived value and panic buying. This can be seen to be consistent with the use of regret theory to portray the relationship between the public's perceived value and panic buying.

Secondly, in this study, the public was influenced by their own perceived value, but also by the surrounding information environment, the social environment, and the rumors spreading environment. This is consistent with Li et al.'s [29] findings that perceived susceptibility and perceived severity of a pandemic event, as well as social influence and social norms, can stimulate consumers' perceptions of scarcity and affective response, which in turn leads to the impulsive decision of panic buying.

Finally, regardless of the public's participation or non-participation in the panic buying, the government prefers to refute the rumors in order to manage the trend of events in time to keep the damage to a minimum. If the public's needs have been met over a period of time, there is less trust in rumors, and government action is perceived as timely, effective, and trustworthy. The most stable equilibrium between the public and the government will be that the public does not participate in panic buying, and the government refutes the rumors.

## 5. Conclusions and Limitations

*5.1. Conclusions*

This paper applies regret theory to describe the public's perceived value of panic buying items in panic buying events, constructs an evolutionary game model with the government and the public as the main participants, and analyzes the equilibrium points of the game and the government's rumors-refuting strategies at each stage of panic buying events, and a simulation was also conducted to analyze the public's rush to buy masks during COVID-19. It examines the impact of the government disinformation strategies and public critical ability on public panic buying behavior. The following conclusions are drawn from the study:

(1) In order to effectively prevent and control the occurrence of panic buying behavior, the government should strengthen real-time monitoring of the rumors spreading situation and the public panic buying signs in the germination period of the epidemic and make adequate preparations for timely policy adjustment. During the intensification period, the government should actively perform its duties, adjust product production, and balance product prices. In the mitigation period, the government should pay attention to repeatedly refuting the rumors and improving the intensity of refuting the rumors, strengthening the openness and transparency of information, reducing the public's trust in rumors, and advocating reasonable public consumption. In the regression period, the government should continue to promote refuting the rumors, timely notify the market supply of items and the disposal progress of panic buying behavior, enhance the public's awareness of loss avoidance, and inhibit the occurrence of panic buying behavior.

(2)  In the group panic buying events, if the government does not take the strategy of refuting rumors, the public will have panic and blind obedience due to their poor self-cognition ability and unsatisfied information needs, which will lead to the panic buying events and interfere with the normal social and market order. If the government adopts the strategy of refuting rumors, it will have a significant impact on the volume of panic buying by the public, enhance the public's critical awareness, and help the public to keep rational and make the right choice. Therefore, the implementation of government strategies will significantly affect the development of group panic buying events. The government should formulate relevant intervention measures at appropriate intervention opportunities to control the impact degree and scope of group buying events.

(3)  If the basic epidemic prevention needs of the majority of the public can be guaranteed, the impact scope of rumors spreading will be reduced, and group panic buying events will be more effectively controlled. Therefore, the government should actively meet the public's demand for epidemic prevention, timely release the price and supply of epidemic prevention products and other information, quickly promote the resumption of work and production of relevant enterprises and effectively ensure the basic needs of the public.

*5.2. Contributions*

(1)  In a theoretical sense, it combines regret theory with an evolutionary game model to study the influence of the public's perceived value on panic buying behavior, describes the characteristics of each stage of the rumors spreading process and the evolution process of panic buying events, and introduces the variable of public critical ability to measure the public's panic buying willingness. This study provides a new theoretical perspective on the study of panic buying behavior, which can be considered from the public's own perspective by considering their regret-joy level to judge their behavioral decisions. In addition, this study extends the application area of regret theory, enriches the study of the factors influencing panic buying behavior, and makes the integration of theory and model more reasonable.

(2)  In a practical sense, this study takes the COVID-19 epidemic as a background, and the study of its panic buying behavior will provide a reference for the establishment and effective implementation of the mechanism to guide public opinion on group panic buying behavior under emergencies and will help to further enhance the government's crisis management capability, strengthen its comprehensive capacity building, actively guide the public to rational consumption, and promote the study of the governance of group panic buying behavior in major public health emergencies.

*5.3. Limitations and Future Research*

There are still some limitations to this study. Firstly, this study only discussed the panic buying events from two participants, the public and the government, without considering the possible influence of other participants, such as the media. Future research could explore the behavioral decisions of multiple participants in the panic buying events by constructing a multi-party game model. Secondly, this study did not fully consider the factors that influence the panic buying events, such as participants' emotions, income levels, media pressure, and other factors. Future research could consider both intrinsic and extrinsic factors to comprehensively analyze the influencing factors of panic buying events. Finally, this study only theoretically explored the governance process of panic buying events and did not fundamentally provide a mechanism for safeguarding snapped items. Future research could establish a rapid response mechanism for the production, supply, and distribution of specific items and explore the governance of panic buying events from the perspective of the supply chain of snapped items.

**Author Contributions:** Conceptualization, Q.-H.M., J.-X.H. and P.-Z.X.; Writing—original draft, Q.-H.M., J.-X.H. and P.-Z.X.; Writing—review and editing, Q.-H.M. All authors have read and agreed to the published version of the manuscript.

**Funding:** This research received specific grant from Hebei Provincial Department of Science and Technology (215576116D), Qinhuangdao Science and Technology Bureau (202005A068), and Key Research Base of Humanities and Social Sciences in Higher Education Institutions of Hebei Province (JJ2109).

**Institutional Review Board Statement:** Not applicable.

**Informed Consent Statement:** Not applicable.

**Data Availability Statement:** Not applicable.

**Acknowledgments:** This work was partially supported by the Soft Science Research Special Project of the S&T Program of Hebei (215576116D), the Science and Technology Research and Development Plan of Qinhuangdao City (202005A068), and the Key Research Base Project of Humanities and Social Sciences in Higher Education Institutions of Hebei Province (JJ2109).

**Conflicts of Interest:** The authors declare no conflict of interest.

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
