# Peer review of "Dynamic Impact of the Perceived Value of Public on Panic Buying Behavior during COVID-19"

_sustainability, doi:10.3390/su14094874_

Round 1

Reviewer 1 Report

1. Introduction - in the introduction I would add information which products are considered as "panic buying items". I suppose that different types of products may generate different patterns of consumer behavior in conditions of uncertainty, spreading rumors and crisis situations. Author mentioned aboaut „emergency goods”. How are emergency goods defined? Food, medicine, fuel, cleaning and hygiene products are also taken into account?

2. Model Building, line 163-170 - the description is chaotic. I sugest writing it in points.

3. A mathematical theoretical model including variants of publick participation and no participation in panic buying and government refutation and no refutation the rumors was prepared, model equilibrium points were derived, simulations were prepared. The model is universal. Is it possible to prepare a short case study on a real example from any Chinese city? It could be a descriptive model. A Chinese example would be extremely valuable due to China's extensive experience in dealing with situations such as CoVID19. I would highly encourage the Authors to do this.

Author Response

请参阅附件。

Reviewer 2 Report

Dear Authors,

I found the paper interesting. The novelty is evident and the methodology and model simulated are valid and sufficiently described. Despite the above, i have a several concerns as follows:

-This is not an empirical work as many assumptions have been made for the model to work. Several theories and concepts have been used to justify the assumptions made. Although they are logical and coherent, at times i have an impression that some assumptions weak and rather 'forceful' to be accepted

-There are many terms that need to be clarified or defined or some background need to be sufficiently provided before they appeared as part of the objective such as stages in panic buying, public or government payment? some contextual background should be provided.

-In the literature, authors may want to talk about opportunity cost they may face if they may such a decision. and cost and benefit analysis may be indirectly involved. Some of these terms can be used interchangeably with the perceived value or the utility gained from such panic buying, critical ability of public and many more.

Line 81-82: This may probably link to the gain and loss (benefits and costs). When consumers feel gains, they feel joyful while they feel remorseful or regret when they probably made a decision that loss is greater than utility/benefits. See prisoner dilemma or game theory (see the following references to strengthen the literature): These can also be used in other parts of the study especially involving the improvements that should be undertaken by the government.

  1. Ling, G. H. T., & Ho, C. M. C. (2020). Effects of the Coronavirus (COVID-19) Pandemic on Social Behaviours: From a Social Dilemma Perspective. Technium Social Sciences Journal, 7(1), 312-320.
  2. Johnson, T., Dawes, C., Fowler, J., & Smirnov, O. (2020). Slowing COVID-19 transmission as a social dilemma: Lessons for government officials from interdisciplinary research on cooperation. Journal of Behavioral Public Administration3(1).

-Hypothesis 3 and 5 are not clear. Kindly explain

-How was the simulation analysis (see section 3.5) conducted?

Another major concern here in this study is that a critical discussion is lacking as to how the findings or simulation model is in line with or contrasting the existing theory of regret and other related theories of panic buying.

In the conclusion or discussion, 

Authors can also summarise what the most stable equilibrium point for both public and the government (what strategies will be dominant?)

One thing for sure is that no matter what position or choice the other party has taken, the government should always refute rumors in all stages of panic buying. I think it would be more meaningful, based on stages and all conditions stated, what dominant strategy should be involved and are considered as equilibrium.

Please state clearly contribution of this paper especially in both theoretical and methodological sense, although policy implication is quite clear. Limitation of the study and future research should be discussed more thoroughly.

See other issues highlighted in the file attached. Please recheck the language as some typos and hanging or unclear sentences appear throughout.

Author Response

请参阅附件。

Reviewer 3 Report

You need to improve the discussion and conclusion

Round 2

Reviewer 2 Report

The paper has been significantly improved; therefore, this paper can be accepted.